# Global Healthcare Needs Related to COVID-19: An Evidence Map of the First Year of the Pandemic

**DOI:** 10.3390/ijerph191610332

**Published:** 2022-08-19

**Authors:** Mariana Aparicio Betancourt, Andrea Duarte-Díaz, Helena Vall-Roqué, Laura Seils, Carola Orrego, Lilisbeth Perestelo-Pérez, Jaime Barrio-Cortes, María Teresa Beca-Martínez, Almudena Molina Serrano, Carlos Jesús Bermejo-Caja, Ana Isabel González-González

**Affiliations:** 1Avedis Donabedian Research Institute (FAD), 08037 Barcelona, Spain; 2Faculty of Medicine, Universitat Autònoma de Barcelona (UAB), 08193 Barcelona, Spain; 3Canary Islands Health Research Institute Foundation (FIISC), 38109 El Rosario, Spain; 4Network for Research on Chronicity, Primary Care, and Health Promotion (RICAPPS), 28029 Madrid, Spain; 5Evaluation Unit (SESCS), Canary Islands Health Service (SCS), 38109 El Rosario, Spain; 6Instituto de Investigación Sanitaria Gregorio Marañón (IiSGM), 28007 Madrid, Spain; 7Fundación para la Investigación e Innovación Biosanitaria en Atención Primaria, 28003 Madrid, Spain; 8Servicio de Medicina Preventiva del Complejo Hospitalario de Toledo, 45004 Toledo, Spain; 9Hermanas Hospitalarias, 28939 Madrid, Spain; 10Unidad de Apoyo Técnico Dirección Técnica de Sistemas de información, Gerencia Asistencial Atención Primaria, Servicio Madrileño de Salud, 28035 Madrid, Spain; 11Departamento de Enfermería. Universidad Autónoma de Madrid, 28034 Madrid, Spain; 12Unidad de Innovación y Proyectos Internacionales, Dirección General de Investigación, Docencia y Documentación, Consejería de Sanidad, 28034 Madrid, Spain; 13Institute of General Practice, Goethe University, 60323 Frankfurt, Germany

**Keywords:** COVID-19, pandemic, needs assessment, healthcare needs, healthcare professionals’ needs, patients’ needs, family members’ needs, evidence map, systematic review

## Abstract

The COVID-19 pandemic has exposed gaps and areas of need in health systems worldwide. This work aims to map the evidence on COVID-19-related healthcare needs of adult patients, their family members, and the professionals involved in their care during the first year of the pandemic. We searched the databases MEDLINE, Embase, and Web of Science. Two reviewers independently screened titles and abstracts and assessed full texts for eligibility. Disagreements were resolved by consensus. Descriptive data were extracted and inductive qualitative content analysis was used to generate codes and derive overarching themes. Thirty-six studies met inclusion criteria, with the majority reporting needs from the perspective of professionals (35/36). Professionals’ needs were grouped into three main clusters (basic, occupational, and psycho-socio-emotional needs); patients’ needs into four (basic, healthcare, psycho-socio-emotional, and other support needs); and family members’ needs into two (psycho-socio-emotional and communication needs). Transversal needs across subgroups were also identified and grouped into three main clusters (public safety, information and communication, and coordination and support needs). This evidence map provides valuable insight on COVID-19-related healthcare needs. More research is needed to assess first-person perspectives of patients and their families, examine whether needs differ by country or region, and evaluate how needs have evolved over time.

## 1. Introduction

The first cases of the new coronavirus infectious disease (COVID-19) were reported in late-December 2019 in Wuhan, China [1]. Due to the alarming levels of spread and severity of the disease, the World Health Organization (WHO) declared COVID-19 as a pandemic shortly after, in March 2020 [2]. The pandemic triggered a global health, social, economic, and human crisis that has exposed gaps and areas of need in systems of care [3].

Since its emergence, COVID-19 has tested the resilience and preparedness of the health systems worldwide. They were overall unprepared to respond to a public health emergency such as COVID-19, and due to the unprecedented nature of the novel coronavirus, had insufficient time and resources to adapt, facing the pandemic with limited effective resources available [4,5,6,7,8]. Limited or inadequate medicines, equipment, personnel, and facilities were reported [9,10,11]. Appropriate service delivery was further compounded by ineffective public health policies and initiatives [12], financial instability, and health information and communication challenges [3].

The significant impact on care systems particularly affected the workforce, service users, and their family members. The health workforce, the driving force of health care supply, struggled with high levels of mental and physical distress [13,14]. They faced enormous work overloads, financial instability, increased risks of infection transmission in part due to insufficient personal protective equipment (PPE), and overall exhaustion [13,14,15]. Moreover, healthcare professionals that were redeployed because of staff shortages faced additional challenges by working in unfamiliar settings [16,17]. Introducing public health safety measures, such as social distancing, stay-at-home-orders, or school and childcare closures, further exacerbated difficulties for professionals, as well as impacted patients and their families [18].

The saturation of the health systems led to a decrease in the quality of services [19,20] and person-centered care [21]. Patients encountered barriers to accessing health services [21,22], communication difficulties with healthcare providers [23], and lack of social support, among other difficulties [22,24]. COVID-19 patients have had pronounced healthcare needs, with approximately half of them needing physical, cognitive, and psychological follow-up care post-acute symptoms’ resolution [25,26,27]. On the other hand, non-COVID patients including people with noncommunicable diseases, such as cancer and cardiovascular disease, experienced disrupted care in general due to a health system responding to the acute pandemic situation [18]. In fact, cessation of essential services to non-COVID patients, and postponing or avoiding seeking medical care contributed to higher morbidity and mortality rates [28]. Besides, previous research has demonstrated the pandemic has increased care disparities, making populations at high social risk, such as underrepresented minorities, more vulnerable to COVID-19 [29]. The growing use of telehealth further reinforced health inequities for marginalized populations who have limited access to technology and financial resources [22]. In turn, families of service users have also been deeply impacted by the COVID-19 pandemic as they have also faced communication, financial, and other social challenges, as well as high psychological distress [30].

The early phase of the COVID-19 pandemic was particularly challenging as the health workforce, patients, and their families were subject to the unpreparedness of health systems and a high degree of uncertainty. To date, no structured attempt has been made to summarize the state of research on COVID-19-related healthcare needs during the first year of the pandemic in these population subgroups. Given the magnitude and the profound impact of the COVID-19 pandemic, and the significance of addressing these needs, mapping the evidence during the first year of the pandemic by conducting a systematic search of existing knowledge in the field is warranted.

### 1.1. Evidence Mapping

Evidence mapping is an emerging method for reviewing and synthetizing evidence in a reproducible and transparent manner [31]. It was first developed and used in the social sciences [32,33], and it has since been widely used in other fields including health sciences (e.g., patient preferences [34], central nervous system injury [35], environmental management [36]). More specifically, evidence mapping is a systematic search of a broad subject area used to identify clusters and gaps in knowledge and/or future research needs that presents results in a user-friendly product, such as a visual or a searchable database [37]. This framework does not require a critical appraisal of included studies but instead aims to assess what type of research has been conducted, what settings have been evaluated, and what methods have been used. Qualitative or quantitative analyses are not always performed; alternatively, the synthesis is limited to a narrative description of the state of knowledge [36]. In sum, evidence mapping is particularly useful as an exploratory, comprehensive evidence synthesis method of a broad topic to provide an overview of the evidence base in an easily digestible format.

### 1.2. Objectives

The present evidence map (EM) aims to (i) describe key characteristics of research on healthcare-related needs associated with COVID-19 from the perspective of adult patients, their family members, and the professionals involved in their care during the first year of the pandemic, and (ii) identify knowledge clusters and gaps to inform future research, clinical practice, and policymaking. This work will provide us with a thorough overview of research on COVID-19-related healthcare needs in the three subgroups of the population most directly affected by the pandemic. This EM also seeks to inform an ongoing online cross-sectional survey study exploring Spanish healthcare and related needs during the care pathway of people with a history of COVID-19, based on the experience of patients and the professionals involved in their care (direct and indirect service providers), in primary care, hospital, and elderly residential care settings.

## 2. Materials and Methods

### 2.1. Protocol Registration and Guideline

A protocol was registered in Open Science Framework (OSF) on 25 January 2021, prior to commencing the screening process (OSF, DOI 10.17605/OSF.IO/3PEYT). Our evidence map adheres to the Preferred Reporting Items for Systematic Reviews and Meta-Analyses extension for Scoping Reviews (PRISMA-ScR) checklist [38] when possible (Appendix A).

### 2.2. Literature Search

The final search strategy was developed in collaboration with an expert librarian. We conducted an electronic search of three databases, MEDLINE, Embase, and Web of Science, from 2019 to 14 January 2021, for publications related to our objectives by using Medical Subject Headings (MeSH) and keywords covering COVID-19, needs assessment, and healthcare needs. Searches were adapted to each database as needed (Appendix A).

### 2.3. Eligibility Criteria

We identified primary work addressing healthcare and related needs associated with COVID-19 from the perspective of adult patients, their family members, and the professionals involved in their care during the first year of the pandemic. For the purposes of our review, “need” is defined as a lack or scarcity of something that is considered essential. A “healthcare need” may be related to treatment, control, management, or prevention, and care or follow-up care [39]. Related healthcare needs may include social care needs or other. Overall, identified needs may be related to the coronavirus disease itself and its consequences, as well as the relationship with relatives and the work or leisure environment, the relationship with professionals, and the relationship with the healthcare system as an entity differentiated from the people who work in it. “Family” is defined as any group of persons who are related biologically, emotionally, or legally such as siblings, parents, spouses, hired caregivers, significant others, and friends (see Omole et al., 2011 for a similar definition of “family” [40]). The terms “family members” and “relatives” are used interchangeably. “Professionals” are defined as any healthcare worker who delivers care and services either directly (e.g., nurses, physicians, psychologists, social workers) or indirectly (e.g., medical waste handlers, managers, security guards, etc.). There were no restrictions on the type of article (e.g., letters), study design, or study setting (country, healthcare, or social context). Table 1 details the inclusion and exclusion criteria.

Two reviewers (MAB, ADD) independently piloted the eligibility criteria on a sample of 50 articles with the aim of achieving 80% agreement. Given a 92% agreement was reached, the eligibility criteria remained unchanged. Two additional reviewers (HVR, ABR) were subsequently trained on the eligibility criteria to provide support during the screening process and obtained at least an 80% agreement.

### 2.4. Study Selection

Metadata of all identified articles were uploaded to Rayyan, a free web application designed to help authors conduct systematic reviews [41]. Two reviewers (MAB, ADD, HVR, ABR) independently screened titles and abstracts for their potential inclusion against the eligibility criteria, and MAB and ADD subsequently screened the full text of all included articles to determine eligibility. Any disagreement was resolved by consensus.

### 2.5. Data Extraction

The data extraction form was piloted by three researchers (MAB, ADD, HVR) independently on five of the included articles and refined prior to extraction. Data were subsequently extracted by one reviewer (ADD) and cross-checked by at least one additional reviewer (MAB, HVR). The following data were extracted: (i) article metadata including citation, publication type (e.g., review, letter), and language of publication, (ii) study characteristics including geographical area, study aim, study setting (e.g., hospital, primary care), study design (e.g., qualitative, quantitative) and methods used to identify needs, and point of contact (i.e., data collection dates and pandemic outbreak phase details), and (iii) population characteristics (e.g., sample size, type of population (participants and population whose needs were identified relevant to our objectives), age, gender).

### 2.6. Data Coding and Synthesis

We conducted inductive qualitative content analysis. Qualitative content analysis is a descriptive and interpretative, iterative process in which codes are generated and counted and patterns in the data are examined [42,43]. Codes were generated based on careful readings of the data, according to the types of needs identified. Data coding was piloted by three authors (MAB, ADD, HVR) independently on five of the included articles to develop and refine the coding framework. Data coding was completed using the NVivo qualitative research data analysis software for Mac (Release 1.4.1, © QSR International Pty Ltd.). The full text of all included articles was first read to determine eligibility and re-read for the purpose of assigning codes. Articles were re-read more times as needed to review and refine the codes individually and during the process of consensus, and to group conceptually similar codes into overarching themes. The first ten articles were independently coded by two reviewers (MAB, ADD), after which codes and emerging themes were discussed to obtain consensus and further refine the coding framework. The remaining data were subsequently coded by one reviewer (ADD) and cross-checked by a second reviewer (MAB). All codes and emerging themes were iteratively discussed and refined until consensus was reached. Data were narratively synthetized and tables and figures were created to summarize and visually depict the data.

## 3. Results

### 3.1. Literature Search and Selection Process

A total of 1626 records were identified from database searching. After removing duplicates, 1336 titles and abstracts were screened, of which 145 remained. A total of 143 full texts were retrieved and assessed for eligibility, of which 36 studies met inclusion criteria (Figure 1). The excluded studies and reasons for exclusion can be found in Appendix A.

### 3.2. Key Characteristics of the Included Studies

The majority of the included studies were published as original research (78%). Studies were conducted in multiple geographical locations: Asia (33%) was the most represented location, followed by Europe (28%), and North America (22%), while only a few studies were performed in Australia (6%), Africa (3%), or represented multiple countries (8%). Healthcare needs were most frequently assessed within hospital settings (44%), followed by studies encompassing professionals in multiple settings (39%). Most of the studies used either qualitative or quantitative methodology for data collection, and fewer studies used mixed methodologies. The majority of needs were identified by professionals, while only one study directly incorporated patient perspectives [45]. No study assessed the first-person perspectives of patients’ relatives. All studies were published in English. Table 2 summarizes the key characteristics of the studies included in this evidence map. Further details of the included studies are presented in Appendix A.

### 3.3. Type of Needs and Evidence Clusters

Professionals’ needs were reported by both patients (*n* = 1) and professionals (*n* = 34), and so were the unmet needs of patients (patients: *n* = 1; professionals: *n* = 12). Family members’ needs were reported by professionals only (*n* = 13). A set of public health disease prevention needs that affect the health of the population as a whole including professionals, patients, and family members, referred to as transversal needs, were also identified. Transversal needs were reported by both patients (*n* = 1) and professionals (*n* = 15). Table 3 presents the main evidence clusters and most frequently reported needs by and across study subgroups. A more detailed description of the type of needs identified in the included studies by and across study subgroups can be found in Appendix A.

#### 3.3.1. Professionals’ Needs during the First Year of COVID-19

Professionals’ needs were grouped into three main clusters: basic, occupational, and psycho-socio-emotional (Figure 2).

##### 3.3.1.1. Basic Needs

Sixteen studies (*n* = 16) reported basic needs, related to physical discomfort or distress, rest, living conditions, and diet and hydration [5,13,46,47,48,49,50,51,52,53,54,55,56,57,58,59]. Across the included studies, healthcare professionals highlighted the need to decrease bodily discomfort and physical distress (e.g., tiredness, excessive sweating, headache, breathing difficulties, etc.), mainly caused by prolonged wearing of PPE [5,13,46,47,48,49,50,51,52,53]. A need for adequate rest in order to remain physically and mentally healthy (both during and after shifts) and be able to provide high-quality care was also reported [5,13,46,47,49,50,51,53,57,58,59]. Many physical and mental challenges of working continuously were noted, which were further exacerbated by sleeping difficulties. Elevated workloads, high patient–professional ratios, and shifts that did not allow for disconnection or rest contributed to these challenges. Healthcare professionals also pointed out the need for lodging support, to adequately self-isolate after the work shift to avoid infecting others, for professionals working long shifts and who do not live close to the health center, and during sick leave [5,13,54,55,56]. Further, healthcare providers stressed the need to have time and access to healthy meals and hydration during and after shifts, especially when wearing PPE and working long shifts [13,46,47,50,51,55,56].

##### 3.3.1.2. Occupational Needs

A wide range of occupational needs were identified in thirty-four studies (*n* = 34) [5,13,45,46,47,48,49,50,52,53,54,55,56,57,58,59,60,61,62,63,64,65,66,67,68,69,70,71,72,73,74,75,76,77], which were grouped into the following seven sub-themes: funding and resources, coordination, information and communication, recognition and support, training, occupational health and safety, and working conditions.

**Figure 2 ijerph-19-10332-f002:**
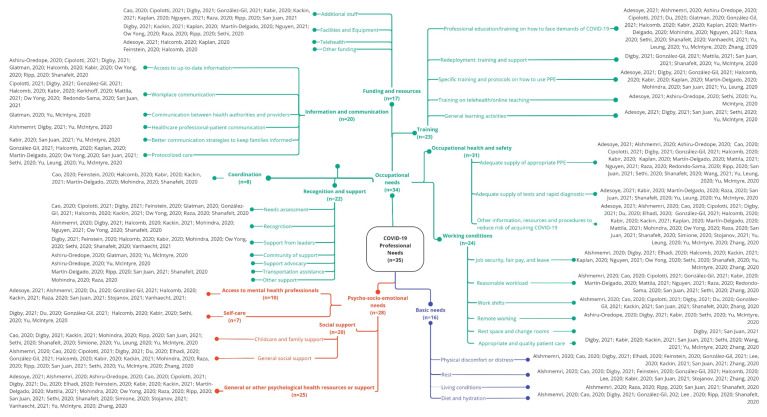
Healthcare professionals’ needs during the first year of the COVID-19 pandemic [5,13,45,46,47,48,49,50,51,52,53,54,55,56,57,58,59,60,61,62,63,64,65,66,67,68,69,70,71,72,73,74,75,76,77].

Funding and Resources (*n* = 17)

An overall need for financial support and additional resources for health services was identified. Professionals highlighted the need for sufficient and adequately trained staff, including the need for additional nurses, doctors, Intensive Care Unit (ICU) workers, mental health staff, and other non-healthcare support staff. This need was related to increased patient volumes, unwillingness to provide services that would place professionals at higher risk of contagion, and sick leave as a high rate of sick leave was observed when more patients with COVID-19 began to emerge [5,46,47,50,52,54,55,58,66,72,73]. Additionally, adequate facilities, equipment (other than PPE), and resource management to meet changing demands were needed, including dedicated staff areas and single rooms, dedicated patient areas for quarantine, dedicated air-conditioning and circulation systems, and sufficient supplies, medications, and beds [47,52,54,55,60,61,62,72,73]. In regard to telehealth services, the following needs were identified: government support, and support establishing services across settings including specific funding to facilitate the transition and reserve in-person visits for patients needing physical attention [57,63,73]. Other financial support and resources for health services were identified, such as the ability for professionals to provide and bill for services different from the regular services, and overall financial support through public–private donations [49,57].

Coordination (*n* = 8)

A need to improve coordination activities was identified. Professionals perceived a lack of task coordination and hoped there could be a more specific task division, as redeployment processes and changes in the working units made it difficult to work with different and changing team members. A clear chain of command in the management and execution of plans, in-person consultation with physicians, and empowerment to delegate tasks and defer less important and time-sensitive activities were also identified as needs [46,49,52,56,57,58,60,71].

Information and Communication (*n* = 20)

Professionals highlighted the need for support concerning the provision of concise, timely, up-to-date, and clear information about the pandemic, so they did not have to research everything themselves (e.g., patient management, infection control protocols, local information, noncontradictory information, etc.) [47,55,56,57,58,61,64,65,66,68]. The need for improved intra- and inter-professional communication was also reported by both patients and professionals, including improved communication between colleagues, team members, and the different services [5,45,47,50,57,58,61,66,67,68]. Accordingly, professionals requested more time to meet with supervisors and colleagues, and staff meetings to discuss as a group the best plan moving forward. There was also a need to improve communication between health authorities and providers to better respond to the emergency situation [65,69], and a need to improve healthcare professional-patient communication, especially when telehealth was used as a substitute for most face-to-face clinic appointments [13,47,69]. Better communication strategies between health authorities, health providers, and families were needed in order to keep families fully informed, particularly when it involved patients in the ICU or when patients were dying [5,58,69]. Finally, the need for standardized, implementable protocols to govern clinical care and logistic planning was noted. Professionals reported the need for additional guidance, such as having access to a go-to person for any clarification on procedures and protocols related to COVID-19 [5,50,57,60,61,62,69,70,73].

Recognition and Support (*n* = 22)

Healthcare professionals emphasized the importance of listening to their perspectives and experiences to identify and rapidly address their needs via tangible actions [46,47,49,50,52,54,56,57,61,65,66]. Professionals wanted to be consulted or be actively involved in decision-making processes, and be regularly updated on the issues that affected them. Moreover, recognition of the vital role played by all professionals during the pandemic emerged as an important unmet need, including frontline and backline professionals [13,47,52,56,57,61,71,72]. The genuine expression of gratitude by the general population is powerful, but professionals stated that this recognition should also come from patients, colleagues, supervisors, managers, organizations, and insurance companies, in the form of real-time support, visible leadership within the workplace and within the profession, and actionable plans [47,49,56,57,58,61,62,71,74]. Accordingly, it was deemed necessary to establish COVID-19 communities of support [64,65,69]. The need for support advocacy for professionals’ safety, education, and delivery of care from national and international healthcare associations was also identified [64,69]. Other support needs such as transportation assistance [5,55,56,60], and overall support from the state, the community, and family, particularly for women, were highlighted [54,71].

Training (*n* = 23)

Training-related needs were one of the main occupational needs identified by healthcare professionals. They perceived a need to receive professional education and training on how to face the demands and complexity of COVID-19 to support the provision of high-quality care to patients during the first year of the pandemic (symptoms, transmission pathway, screening procedures, treatment, patient management, counseling skills, working with vulnerable populations, infection control, etc.) [13,50,53,54,56,57,58,60,62,63,64,65,66,69,70,71,72,73,74,75]. Healthcare professionals also needed to receive specific training on how, when, and where to wear PPEs as they reported fear of not using it properly because of the lack of adequate instructions or practice and frequently changing requirements [5,47,50,57,58,60,63,70,71,73]. We also identified the need for sufficient training and support in preparation for redeployment and during the onboarding process to work in clinical areas that were outside professionals’ usual scope of practice [5,47,50,56,67,69]. Similarly, the transition to online services was a huge challenge for some professionals and for organizations in general, and receiving specific training on telehealth and online teaching was stated as a possible way to overcome this challenge [62,63,64,69]. Some issues identified as barriers to providing virtual care included a lack of familiarity with the different technological systems and the logistics of scheduling and billing for virtual visits. Finally, some studies highlighted the need to minimize COVID-19’s impact on general learning activities including medical education and research, as doctors in training faced uncertainty regarding examinations and courses that had been paused, which in turn could impact the trainees’ preparedness for independent practice [5,47,62,63,69].

Occupational Health and Safety (*n* = 31)

The need for an adequate and timely supply of appropriate PPE, at low or no cost, and ways to preserve existing PPE was one of the most frequently reported needs from the perspective of healthcare professionals (e.g., N95 masks, face shields, alcohol, and different gowns for different departments and healthcare settings) [5,13,46,47,50,54,55,56,57,58,60,62,63,64,66,67,68,69,70,72,73,76]. The unavailability and substandard quality of PPE were indeed one of the major causes of professionals’ frustration and distress. An adequate supply of tests, access to testing through occupational health, and rapid diagnosis for all staff, when necessary, was also needed to ensure the health and safety of professionals [5,54,56,58,60,63,69,70]. Moreover, based on professionals’ constant fear of self-infection (especially those at high risk because of age or health conditions) or spreading the virus to their families, friends, and patients, the need for up-to-date information, resources, and procedures aimed to reduce the risk of becoming infected or being a portal for disease transmission was noted (e.g., accommodation, insurance for all, designated floors, negative pressure airborne infection isolation rooms, etc.) [5,13,46,47,48,50,52,53,54,56,57,58,59,60,61,63,66,67,69,70,71,73,75,77].

Working Conditions (*n =* 24)

Ethical dilemmas emerged due to worsening working conditions. Job security, fair pay and sick leave arose as one of the main needs of healthcare professionals in this subtheme [13,47,48,52,53,56,57,61,62,69,72,73]. They stated the importance of having financial stability and support and a clear understanding of the human resources policies. Reasonable workloads [5,13,46,50,53,54,58,60,62,66,67,68,72] and suitable work shifts [5,13,46,47,50,52,53,56,66,75] were also identified as two important concerns to healthcare workers during the first year of COVID-19. Healthcare workers expressed concerns about the emotional impact of working continuously under challenging circumstances, with frequent changes to processes and procedures, and being unable to provide high-quality care, as the workload increased substantially as staff became ill. They reported decreased concentration after working long hours, in part due to high patient-professional ratios and night shifts that did not allow them to disconnect or rest, amplifying the need to hire additional staff. Access to rest spaces, change rooms, and lunchrooms were also needed to alleviate the strenuous working conditions [5,47].

Professionals who transitioned to work remotely expressed the need to implement high-quality telehealth services, and for organizations and professionals to adjust to the specific conditions of remote working [47,58,62,64,69]. Remote working was a positive experience for some, with fewer interruptions, less commuting time, and more flexibility. In other cases, the home environment was unsuitable for working remotely due to poor internet connection, noisy or distracting housemates (including children), or inadequate space and facilities. Decreased quality of care was also reported for patients without COVID-19, especially those with chronic conditions, during end-of-life care, for care home residents, and for clinical rehabilitation services. Accordingly, healthcare professionals stated the need to provide appropriate and high-quality care even during the peak of the pandemic [5,47,52,53,58,62,69,76].

##### 3.3.1.3. Psycho-Socio-Emotional Needs

Twenty-eight studies (*n =* 28) reported psycho-socio-emotional needs from the perspective of professionals. These were grouped into the following four subthemes: Access to mental health professionals, self-care strategies, social support, and general or other psychological health resources or support [5,13,46,47,48,49,50,52,53,54,55,56,57,58,59,60,61,62,63,64,66,67,69,70,71,74,75,77].

Access to Mental Health Professionals (*n =* 10)

Psychological health was highlighted as a major concern during the first year of the pandemic. In a time of extreme work hours, uncertainty, and intense exposure to critically ill patients, professionals voiced the need to have access to mental health professionals [5,13,50,52,54,57,59,63,74,75]. Therapeutic options needed varied and included individual or group sessions, in-person or online, and short- (e.g., channel or helpline) or long-term support provided by psychologists and therapists in related fields.

Self-Care (n = 7)

Self-care is a “multidimensional, multifaceted process of purposeful engagement in strategies that promote healthy functioning and enhance well-being” [78]. Across the included studies, a need for organizations to encourage professionals’ self-care and for professionals to engage in self-care in order to enhance wellbeing was identified (e.g., limiting exposure to media, psycho-education, expressing feelings, meditation, yoga or other relaxation techniques, exercise, singing, writing or other recreational activities) (Although social support is also perceived as a self-care practice, it was coded separately given emphasis and frequency noted) [47,50,52,57,62,69,75].

Social Support (*n =* 20)

Many staff members had to distance themselves from family and friends due to work (e.g., long work hours, self-imposed or ordered quarantines) or their loved ones’ fears of becoming infected. In a context where the usual social networks and leisure activities used by healthcare workers to de-stress were restricted, the need for other forms of social support and social connection was highlighted (e.g., vent emotions with supportive people, informal peer support groups, virtual educational activities, video chat with family and friends) [5,13,46,47,48,50,52,53,54,55,57,58,62,66,69,71,75]. Healthcare professionals also highlighted the need for childcare and family support, including financial support, as work hours and demands increased and schools and daycare closures occurred [5,46,47,52,55,56,62,69,70,71,77].

General Psychological Health Support or Other Psychological Health Resources (*n =* 25)

During the first year of COVID-19, a mental health crisis began to emerge in healthcare workers. Healthcare professionals worldwide experienced high levels of stress, anxiety, and depression, which directly affected patient care and professionals’ personal and family wellbeing. Hence, most of the included studies deemed the need for other resources and general psychological health support to help healthcare workers cope with emotional symptoms and anticipate mental health needs [5,13,46,47,48,49,52,53,54,55,56,58,59,60,61,62,63,64,66,67,69,71,74,75,77].

#### 3.3.2. Patients’ Needs during the First Year of COVID-19

Patients’ needs were grouped into four main clusters: basic, healthcare, psycho-socio-emotional, and other support needs (Figure 3).

##### 3.3.2.1. Basic Needs

Basic needs, including access to essential goods and resources, and lodging support, were identified in two studies (*n =* 2) [45,68]. As the pandemic spread, patients required support for urgent and regular home deliveries of essential supplies during isolation (e.g., food, PPE, cleaning supplies, hygiene products, supportive care medication) [45,68]. Additionally, patient’s identified the need to access information and services in their preferred language [45]. Finally, a need for lodging support for patient’s to safely isolate also emerged, related to shared living spaces or homelessness [45,68].

##### 3.3.2.2. Healthcare Needs

Patients’ healthcare needs were highlighted in eleven studies (*n =* 11) [5,13,24,45,47,49,54,58,68,69,71]. The need for support obtaining healthcare coverage was essential in those countries without public health services [45,54]. A deep concern emerged because of the unrealistic charges per day COVID-19 patients needed to afford. The need to improve the quality of care for all people during the pandemic was highlighted, in particular for vulnerable populations such as low-income families, patients with underlying conditions, care home residents, elderly, and homeless people [24,58,68]. Accordingly, some studies pointed out the need to receive additional short- and long-term quality healthcare services, such as nutritional, counseling, and rehabilitative care [24,45,71]. Patient safety was another concern, as in some cases, the isolation wards lacked the necessary facilities and equipment. Patients must always come first, and to ensure patient safety, adhering to internationally recognized standard procedures and patient safety strategies is needed [24,54]. Further, the need to improve professional-patient communication was noted [13,24,58]. Building strong connections with patients and providing emotional support could alleviate negative feelings related to isolation, physical problems, and the lack of a definitive treatment. During the first year of COVID-19, the idea that patients could die alone was particularly distressing. In this respect, four studies in different countries highlighted the need to preserve patient rights during end-of-life care and immediately after death [5,47,49,58]. Finally, the need to address other healthcare needs arose, including hospital environmental familiarization, the need for patients to be fully informed of their health status, the need to facilitate low-barrier testing for close contacts, and the need for up-to-date and clear information on COVID-19 and rehabilitation [24,45,69].

##### 3.3.2.3. Psycho-Socio-Emotional Needs

Psycho-socio-emotional needs for patients with COVID-19 were identified in nine studies (*n =* 9) [13,24,45,50,54,58,68,71,76]. As COVID-19 patients might suffer from many mental health concerns (e.g., death anxiety, social stigma, hopelessness, separation anxiety) during and after the disease, the need for access to mental health professionals was noted [13,24,50,54,68,71]. Additionally, the need for low-barrier social support services, including childcare and social welfare of the patient and their family, to mitigate economic concerns was identified [24,45,68,71,76]. Furthermore, emotional support for COVID-19 patients was highlighted as a primary requirement, including more informal support (e.g., general affection, peer support groups) and also the need to implement screening programs to identify patients in need of psychological counseling [24,45,50,58,71].

##### 3.3.2.4. Other Support Needs

Finally, eight studies (*n =* 8) identified other support needs for COVID-19 patients during the first year of the pandemic [5,24,45,54,58,68,69,76]. First, continuous communication and social interaction with family and friends was stated as one important unmet need [5,24,54,58,68,76]. As humans are social beings, special attention should be paid to patients isolated due to COVID-19 to decrease feelings of homesickness and loneliness during this period. Accordingly, allowing patients to communicate with their families and relatives during isolation and hospitalization was highlighted. Patients’ perspectives are key to providing comprehensive patient care. For this reason, patients’ needs should be continuously assessed, identified, and addressed [24,45]. Given the uncertainties and the changes to healthcare and community resources resulting from policies aimed to manage the pandemic, all patients with COVID-19, but especially vulnerable patients, such as older patients and people with disabilities, would benefit from support advocacy [68,69]. Finally, one study highlighted spiritual care is needed to provide comprehensive care as it gives patients mental peace [24].

#### 3.3.3. Family Members’ Needs during the First Year of COVID-19

Family members’ needs were grouped into two main clusters: psycho-socio-emotional and communication (Figure 4).

##### 3.3.3.1. Psycho-Socio-Emotional Needs

Psycho-socio-emotional needs of patients’ relatives were identified in eight studies (*n =* 8) [5,47,49,50,66,68,71,75]. It was noted that family members of COVID-19 patients needed psychological and/or emotional support during and after the patient’s acute illness, including during the mourning process [49,50,66,68,75]. Some studies also highlighted the need of providing social support to family members, including financial support to mitigate economic concerns, and childcare assistance while infected patients were hospitalized [50,68,71]. Finally, the need for families to accompany dying patients whenever possible was stressed [5,47,49].

##### 3.3.3.2. Communication Needs

Seven studies (*n =* 7) included communication needs [5,54,58,68,69,76]. As during the first year of the pandemic visits were mostly forbidden, there was a need for better communication between health authorities, health professionals, and families to keep families informed [5,58,69]. Families often had to hear difficult-to-understand information about their relatives’ health status over the telephone, rather than in person, and this lack of face-to-face communication made interacting with families of dying patients even more challenging. Moreover, the need for continuous communication (e.g., regular updates of the patients/relatives’ health status) and interaction between family and patients/residents was noted [5,24,54,68,76].

#### 3.3.4. Transversal Needs during the First Year of COVID-19

Transversal needs were grouped into three main clusters: public safety needs, information and communication needs, and other needs (Figure 5).

##### 3.3.4.1. Public Safety Needs

Public safety needs were identified in nine studies (*n =* 9). Needs in this cluster elucidated the importance for the general public to adhere to disease prevention protocols [13,24,62,68,77], need for the government and/or health authorities to provide professionals and the general population with PPE to minimize the spread of the virus [57,68,70], and overall need for more effective public health policies to contain the outbreak (e.g., border control and/or quarantine measures for returning residents and travelers, follow up of asymptomatic cases, more restrictive containment measures) [47,60,70,77].

##### 3.3.4.2. Information and Communication Needs

Information and communication transversal needs were identified in sixteen studies (*n =* 16). To avoid spreading the virus, culturally tailored health-related and COVID-19 education and awareness should be improved in the general public (e.g., principles of disease prevention, awareness of risks, symptoms and linkage to care, instructions about where to get tested) [13,24,45,54,57,60,62,68,70,72]. Additionally, fake news and misinformation should be rapidly clarified [24,47,57,62,68,70]. Moreover, efforts should be undertaken to reduce social stigma [5,24,48,52,54,71]. Social stigma was one of the causes of patients’ anxiety as they feared social rejection, and a major cause of suppression of travel history and incompliance with transmission prevention strategies.

##### 3.3.4.3. Other Needs

Other needs related to coordination and support were identified in six studies (*n =* 6). More specifically, the need for coordinated care between the public and private sectors was noted [45,47,54,70,71], as well as the need to provide support to vulnerable groups to minimize the spread of the virus, including the provision of basic needs (food, housing, access to shower, etc.) [68].

## 4. Discussion

Early on in the pandemic, COVID-19 caused serious repercussions to daily life that were felt across the globe, raising social, economic, political, and global safety concerns. As a first step towards identifying health systems’ limitations to subsequently improve their resilience and preparedness for future global crises, it is important to explore how COVID-19 first affected healthcare workers, patients, and families. To the best of our knowledge, this is the first study to map knowledge clusters and gaps regarding healthcare and related needs of adult patients, their family, and the professionals involved in their care during the first year of the COVID-19 pandemic.

### 4.1. Evidence Clusters

The impact of this pandemic on the physical and mental health of healthcare professionals has been highlighted in several reports since the initial outbreak. A plethora of studies examined the prevalence of mental health symptoms in healthcare professionals facing the pandemic on the front line, reporting moderate and high levels of stress, anxiety, depression, sleep disturbance, and burnout [79,80,81]. In the aftermath of the COVID-19 outbreak, healthcare professionals had to work under extreme pressure and needed to balance conflicting demands related to their own physical and mental health as well as those of colleagues, patients, and family members [54]. The evidence suggests that professionals’ occupational and psycho-socio-emotional needs are intertwined, as clinical well-being is heavily influenced by organizational factors of well-being. Different studies have pointed out several work-related risk factors that may lead to increased occupational stress and emotional exhaustion, including access to or prolonged use of PPE, increased work pressure, lack of social support from colleagues and supervisors, and prolonged work hours [82,83]. A perceived risk or fear of COVID-19 can also contribute to mental health problems [84]. The lack of PPE and the subsequent fear of becoming a portal of transmission were two of the most common causes of frustration and distress in the included studies, which makes this especially relevant. Even when PPE was available, healthcare workers voiced the need for specific training on how, when, and where to wear PPE. Accordingly, recent studies with frontline healthcare workers have suggested that access to high-quality PPE and in-person training on how to use them, should be the minimum standard to prevent transmission and ensure safety. Greater confidence in using PPE also correlates with an improved workplace culture [85]. Our results suggest that mental health support strategies and easy-to-implement, flexible, and effective mental health interventions for healthcare professionals should be incorporated into pandemic preparedness plans, while improving working conditions and developing mechanisms to increase occupational safety in these high-risk clinical settings. To date, most interventions have mainly focused on the individual level, but proactive organizational approaches are also needed and could be less stigmatizing [83], especially when the threshold for healthcare professionals to seek help might be higher compared to other population groups [67]. Finally, to develop effective approaches to support healthcare providers, it is essential for providers to be involved in the decision-making process.

In regard to patients, ensuring that all patients have access to essential goods, services, lodging support and information should be an aspect included in preparedness plans. Several articles highlight the need to improve the healthcare service quality (e.g., improve hospital facilities and equipment) in order to guarantee that all patients receive high-quality care. This stresses the importance of providing universal access to health services to ensure that all patients get adequate treatment. Improving communication between patients and healthcare professionals is another key aspect that needs to be considered: the current healthcare environment affords little time with each patient, and this can impede effective patient-professional communication. Patient-centered interviewing, caring communication skills, and shared decision making are strategies known to improve patient-professional communication [86]. Given the serious nature of the COVID-19 outbreak, it is not surprising that a high proportion of patients experienced depression, anxiety, and post-traumatic symptoms [81,87]. Therefore, early detection and appropriate treatment of mental health symptoms are required in patients with COVID-19, both during and after acute symptom resolution, in order to reintegrate with society and family life [88].

COVID-19 has also had a profound impact on patients’ relatives. Deleterious effects on the quality of life of the survivors’ family members have been reported, with partners being most impacted. According to Shat et al. [89], many family members reported being worried and frustrated, and experienced sadness and difficulties in caring for their loved ones. Similarly, Kentish-Barnes et al. [90] found that having a loved one who died or was close to death and reporting poor communication with the healthcare professionals are risk factors for psychological burden. In line with other work, we identified several psycho-socio-emotional and communication needs. The establishment of family-centered services to provide support to patients with COVID-19 and their family members is therefore a key consideration in the future management of this pandemic and possibly other serious threats.

### 4.2. Evidence Gaps

Some important evidence gaps have been highlighted in our work. First, more studies focusing on first-person perspectives from patients and family members are needed. However, based on the needs identified, moving forward, organizations should plan to implement comprehensive strategies of care that include specific family-centered guidelines for crisis management and respect the voices of patients and their families [90]. Our results indicated the need for low-cost and low-barrier access to support for patients and families, including: basic needs (lodging support, access to essential goods, and access to information and services in patients’ preferred language), healthcare coverage, mental health and social support services, and other. Similarly, more effective public health initiatives are needed, particularly targeting vulnerable populations. As stated by Chua et al. [91], preparedness for future pandemics and global health crisis requires adopting more inclusive responses that protect all individuals, including the elderly, low-income, homeless, and disabled people. Health literacy is key to empowering individuals and their communities and can be used as a tool to promote public safety during health emergencies.

Furthermore, addressing the barriers to implementing organizational changes to support wellbeing is needed, as addressing the wellbeing of healthcare professionals requires a whole system participatory approach [5]. Our work demonstrated organizational factors such as shift planning, workload, communication within the workplace, visible leadership, involvement in decision-making processes, etc., need to be addressed in addition to providing psycho-socio-emotional support.

We identified a further research gap in a lack of studies performed beyond Asia, Europe, and North America, especially in low-income countries. A more comprehensive understanding of the healthcare and social needs arising from the first COVID-19 outbreaks worldwide will require more insights from Africa and other underrepresented regions, such as Oceania and South and Central America.

Finally, as we only focused on the first year of the COVID-19 pandemic, more research is required to assess how needs evolved over time and which unforeseen needs arose as the outbreak progressed (e.g., rehabilitation needs, vaccination-related issues). The identified needs will need to be prioritized and addressed.

### 4.3. Implications for Practice

A wide variety of improvement opportunities emerged from this study. Policymakers, managerial staff, and clinicians can learn from this mapping exercise by promoting better preparedness and resilience at multiple levels. Researchers, on the other hand, should test and evaluate the effectiveness of interventions tailored to meet the identified needs, at different levels:Macro-level Interventions

Promote channels of information and official communication on a regular basis that is up-to-date, concise and clear; facilitate staff hiring processes or streamline external subcontracting of services.

Meso-level Interventions

Setting up coordination teams to channel and structure training and information needs, so that high-quality care can be delivered while ensuring that decision making involves multiple stakeholders. Develop an emergency response plan adapted to pandemics, including specific actions, roles, and responsibilities defined with a multisectoral approach and integration of primary and residential care. These plans should include adequate facilities (e.g., isolation rooms), resource management protocols and other easy-to-implement protocols; material resources, support, and training for professionals; and accompaniment of patients at home. Attention to the mental health of professionals and patients should be ensured, favoring access to mental health professionals, evidence-based self-care practices and social support, including peer teams (similar to already tested strategies for second victims of adverse events) or self-help groups adapted to pandemic status and local idiosyncrasy/culture.

Clinical level Interventions

Emphasis on person and kin-centered care, facilitating clear and reliable communication with patients and families, establishing liaison professionals or volunteers, and streamlining the use of new technologies to maintain connections and social relationships. Establishment of specific end of life care channels to ensure support and care for COVID and non-COVID pathologies.

### 4.4. Strengths and Limitations

A major strength of this study is that we used a broad-based search strategy and selection criteria, with no restrictions on the type of article, study design, or setting. Moreover, the qualitative content analysis employed allows for rich description while able to summarize the evidence via frequency count. However, this evidence map also presents some limitations. Our findings only included published peer-reviewed work, which has been limited by the pressing needs caused by COVID-19, leaving out potentially non-published but relevant work. A search for backward and forward citations was not performed, nor did we include gray literature. A quality analysis of the included studies was not conducted, although this is not a requirement for evidence maps. In addition, some of the needs and gaps identified during the first year of the pandemic have already largely been addressed, such as the need for an adequate supply of appropriate PPE for professionals [92]. Nevertheless, this information will be useful in facilitating the preparation of health systems for future pandemics. Other identified needs and gaps, however, are still current and pressing, such as the need to improve access to mental health professionals; efforts should be undertaken to prioritize and address them. While our focus was on the early phases of the COVID-19 pandemic, further research should focus on more recent needs (e.g., those related to vaccination and long-term consequences of COVID-19). Moreover, the generalizability of the results is limited as some geographical areas such as Africa, Oceania, and South and Central America are underrepresented. Finally, most of the reviewed evidence was based on healthcare professionals’ viewpoints, with only one study directly incorporating patient experiences and none addressing family members’ first-person perspectives.

## 5. Conclusions

This evidence map provides valuable insight on healthcare-related needs associated with COVID-19 from the perspective of patients and professionals based on the first year of the pandemic, by: describing key COVID-19-related healthcare needs, identifying knowledge clusters and gaps, and highlighting improvement opportunities that may facilitate multilevel preparedness and resilience. This work elucidated the need for more first-person perspectives from patients and family members. Altogether, our results indicate COVID-19 amplified the need to provide person and kin-centered care, and for health systems to better prepare for public health emergencies, and better support the workforce through organizational approaches to wellbeing. Given the profound multifaceted impact of the pandemic, addressing current healthcare-related needs should be prioritized. Further research is needed to identify which of the identified needs are more relevant, develop initiatives to address those needs, and detect barriers to implementation. Additional research is also warranted to assess whether needs differ by country or region, and to evaluate how needs have evolved over time given the magnitude and dynamic nature of the COVID-19 pandemic.

## Figures and Tables

**Figure 1 ijerph-19-10332-f001:**
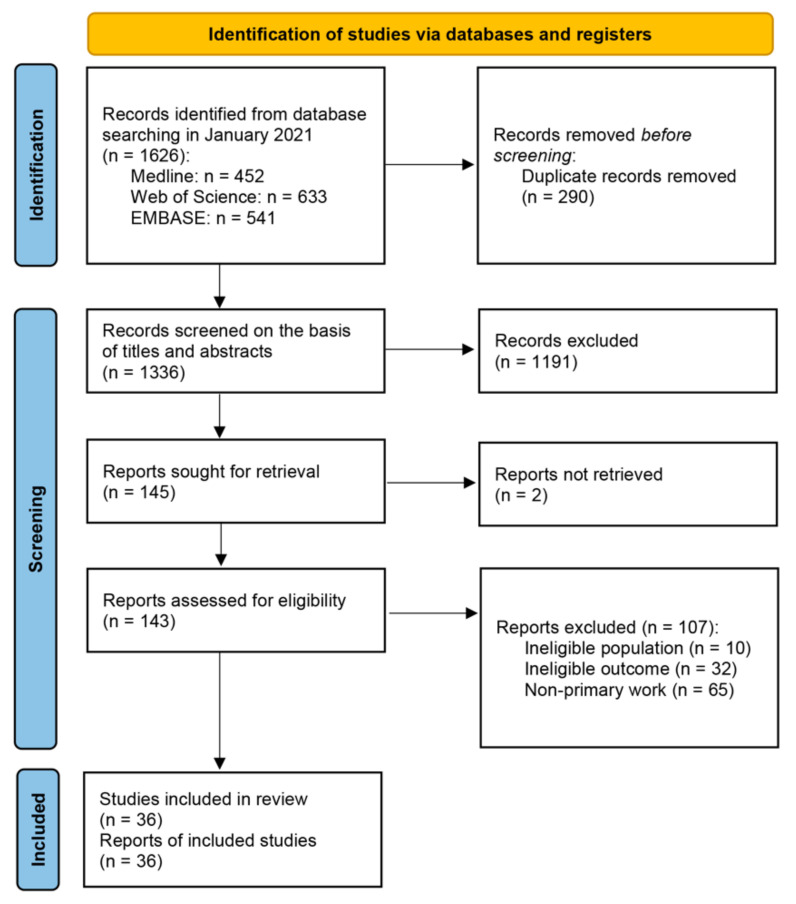
PRISMA flowchart. From: Page et al., (2021) [44].

**Figure 3 ijerph-19-10332-f003:**
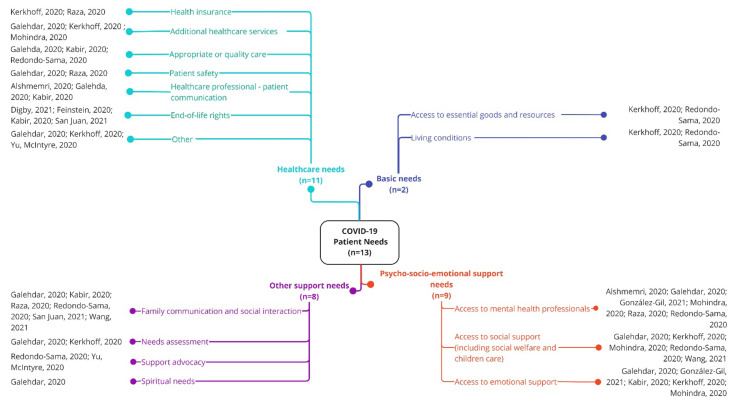
Patients’ needs during the first year of the COVID-19 pandemic [5,13,24,45,47,49,50,54,58,68,69,71,76].

**Figure 4 ijerph-19-10332-f004:**
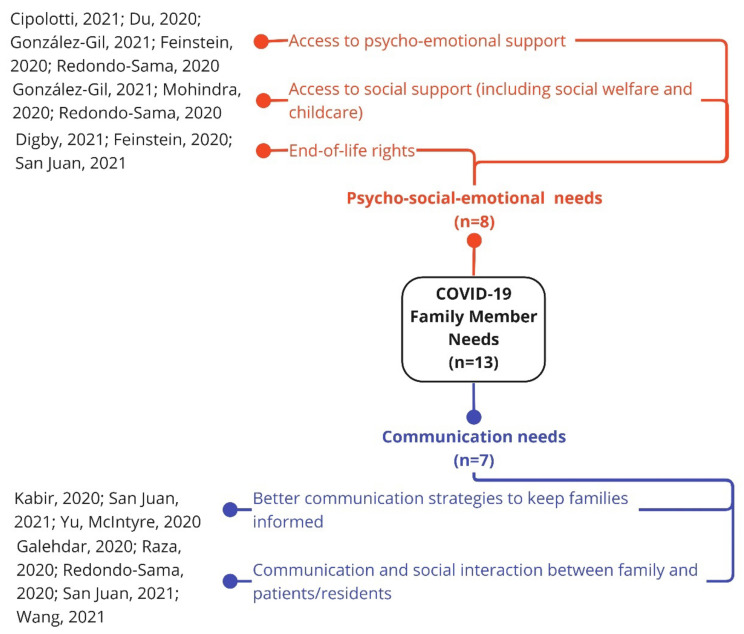
Family members’ needs during the first year of the COVID-19 pandemic [5,24,47,49,50,54,58,66,68,69,71,75,76].

**Figure 5 ijerph-19-10332-f005:**
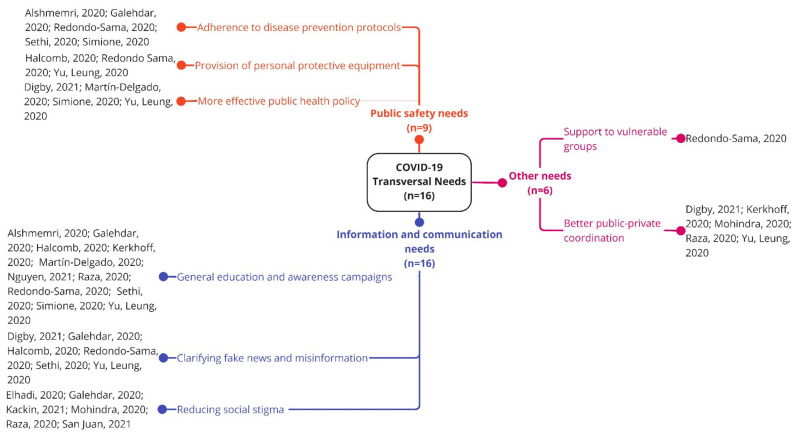
Transversal needs during the first year of the COVID-19 pandemic [5,13,24,45,47,48,52,54,57,60,62,68,70,71,72,77].

**Table 1 ijerph-19-10332-t001:** Inclusion and exclusion criteria.

Inclusion Criteria	Exclusion Criteria
Primary qualitative and/or quantitative work addressing any healthcare-related needs associated with COVID-19 from the perspective of adult patients, their relatives, and the professionals involved in their care.	Addressed needs in the general population, but not specifically in our population subgroups. Focused on exploring the prevalence of a mental health illness or psychological factor, but did not aim to identify healthcare or related needs associated with COVID-19 from the perspective of one of the population subgroups. Published in languages other than those known by the research team: English, Spanish, or Catalan.

**Table 2 ijerph-19-10332-t002:** Summary of key characteristics of included studies.

	Variable	*n* (%)
**Type of publication**		
	Original research	28 (78%)
	Brief report	3 (8%)
	Perspective	2 (6%)
	Letter to the editor/commentary	2 (6%)
	Review	1 (3%)
**Geographical location**		
	Asia	12 (33%)
	Europe	10 (28%)
	North America	8 (22%)
	Multi-country	3 (8%)
	Australia	2 (6%)
	Africa	1 (3%)
**Healthcare setting**		
	Hospitals	16 (44%)
	Multiple settings	14 (39%)
	Elderly residential care	2 (6%)
	Primary care	2 (6%)
	Community-based	1 (3%)
	Unspecified	1 (3%)
**Study design**		
	Quantitative	14 (39%)
	Qualitative	14 (39%)
	Mixed	8 (22%)
**Needs identified by**		
	Professionals	35 (97%)
	Patients	1 (3%)

**Table 3 ijerph-19-10332-t003:** Main evidence clusters and most frequently reported needs by and across study subgroups.

	Professionals	Patients	Family Members	Transversal Needs
Total No. of Studies	35/36 (97%)	13/36 (36%)	13/36 (36%)	16/36 (44%)
Evidence clusters	Basic needs (*n* = 16) Occupational needs (*n* = 34) Psycho-socio-emotional needs (*n* = 28)	Basic needs (*n* = 2) Healthcare needs (*n* = 11) Psycho-socio-emotional needs (*n* = 9) Other support needs (*n* = 8)	Psycho-socio-emotional needs (*n* = 8) Communication needs (*n* = 7)	Public safety needs (*n* = 9) Information and communication needs (*n* = 16) Other needs: Coordination and support (*n* = 6)
Most frequently reported needs	General or other psychological health resources or support (*n* = 25) Adequate supply of appropriate personal protective equipment (*n* = 22) Other information, resources, and procedures to reduce risk of acquiring COVID-19 (*n* = 24)	Access to emotional support (*n* = 5) Family communication and social interaction (*n* = 6) Access to mental health professionals (*n* = 6) Access to social support (including social welfare and childcare) (*n* = 5)	Communication and social interaction between family and patients or residents (*n* = 5) Access to psycho-emotional support (*n* = 5)	General education and awareness campaigns (*n* = 11) Clarifying fake news and misinformation (*n* = 6) Reducing social stigma (*n*= 6)

## Data Availability

Not applicable.

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
