# Peer review of "Global Healthcare Needs Related to COVID-19: An Evidence Map of the First Year of the Pandemic"

_ijerph, 2022, doi:10.3390/ijerph191610332_

Round 1

Reviewer 1 Report

Brief Summary: The objective of this study was to map the evidence on COVID-19-related needs of healthcare professional workers, patients and family member during the forst year of the pandemic. Authors used as a tool for reviewing and synthetizing data the evidence mapping, in order to describe key characteristics of related needs associated with COVID-19 and identify knowledge clusters and gaps to facilitate further research and clinical practice. Authors conclude that this evidence map provides valuable insight on COVID-19-related healthcare and social needs, highlighting though that these needs differ by country, region, and also evolve over time.

Broad comments: This study deals with an extremely important issue during pandemic. The article is well-organized and explores thoroughly all aspects. Separate sections, Introduction, Methodology etc, are well-developed, especially Results. The results are interesting covering all aspects, healthcare workers and patients, less family members. Sound methodology, clear messages, although very lengthy and wordy. To become more attractive to readers and less tiresome, It has to be limited in words and pages.

Specific comments: In detail:

Abstract: Well written

Introduction: Well written. Lines 76-78, sentence “Introducing public health … , and their families alike. [18]” does not match to the previous paragraph

Methodology: methodology is clear and focused. 

Table 1, exclusion criteria, studies in parentheses unable to track. Supplementary material S3 is more than enough.

Results: Results are also well expressed, interesting, and easy to follow and understand. Nevertheless, results are extremely wordy, lengthy and repetitive meanings and concepts are part of it, while it becomes tiresome for the reader. I am not sure if the length of the manuscript is acceptable by the Editor. Figures 2, 3, 4 are vague, probably could also be supplements. 

Discussion: Authors critically and in depth discuss their results and they address the limitations of their study.

Conclusions: clearly stated what this study found.

Thank you for the privilege of reviewing your work

Author Response

Below we provide a point-by-point response to reviewer 1. In addition, attached you will find a document containing the responses to all reviewers:

Reviewer 1

Broad comments: This study deals with an extremely important issue during pandemic. The article is well-organized and explores thoroughly all aspects. Separate sections, Introduction, Methodology etc, are well-developed, especially Results. The results are interesting covering all aspects, healthcare workers and patients, less family members. Sound methodology, clear messages, although very lengthy and wordy. To become more attractive to readers and less tiresome, It has to be limited in words and pages.

We have streamlined the results section based on the reviewer’s recommendations. 

Specific comments: 

Abstract: Well written

Introduction: Well written. Lines 76-78, sentence “Introducing public health … , and their families alike. [18]” does not match to the previous paragraph

We interpreted this comment to mean this sentence needs to be better contextualized, and have rephrased the sentence to address this concern. The paragraph now reads,

“The significant impact on care systems particularly affected the workforce, service users, and their family members. The health workforce, the driving force of health care supply, struggled with high levels of mental and physical distress [13,14]. They faced enormous work overloads, financial instability, increased risks of infection transmission in part due to insufficient personal protective equipment (PPE), and overall exhaustion [13–15]. Moreover, healthcare professionals that were redeployed because of staff shortages faced additional challenges by working in unfamiliar settings [16,17]. Introducing public health safety measures, such as social distancing, stay-at-home-orders, or school and childcare closures further exacerbated difficulties for professionals, as well as impacted patients and their families [18].”

Methodology: methodology is clear and focused. 

Table 1, exclusion criteria, studies in parentheses unable to track. Supplementary material S3 is more than enough.

We have moved this information to Supplementary Material S3: List of excluded studies and reasons for exclusion.

Results: Results are also well expressed, interesting, and easy to follow and understand. Nevertheless, the results are extremely wordy, lengthy and repetitive meanings and concepts are part of it, while it becomes tiresome for the reader. I am not sure if the length of the manuscript is acceptable by the Editor. Figures 2, 3, 4 are vague, probably could also be supplements. 

We have briefly reduced the length of the results section, while largely maintaining the original idea. 

Given Evidence Maps are characterized by incorporating user-friendly products and a visual depiction of the data (Miake-Lye et al., 2016), and the positive feedback provided by another reviewer regarding the usefulness of the visuals, we have kept the evidence maps (figures 2-5) in the main text. To make the maps more specific and useful, we have added the frequency count (i.e., the number of studies) for each Key Theme and Sub-theme, consistent with qualitative content analysis. 

Discussion: Authors critically and in depth discuss their results and they address the limitations of their study.

Conclusions: clearly stated what this study found.

Thank you for the privilege of reviewing your work

Reviewer 2 Report

The paper meets all of the requirements of an “evidence map” as the specified protocol, a systematic search strategy, a systematically applied, clear inclusion and exclusion criteria, and reports all eligible studies (in the supplementary file S4).

Being in the third year of the COVID-19 pandemic (when vaccines are available, changing almost fundamentally the situation along with the healthcare needs, as it is pointed out among the limitations of the study) the importance of the presented data regarding the first year of the pandemic is decreasing, but the work still has a significant contribution to the field.

Some of the identified needs and gaps are no longer valid; there are needs, which are not relevant anymore (personal protective equipment are available now, a wider range of knowledge about the virus is available) and other still stands, but in the chapter 4.3. (Implications for practice) the valid interventional proposals are formulated.

It is a thoroughly documented, very well-written paper.

Row 161: “see also 3” – is not obvious what it refers to

Table 3 – bullets are not necessary, and the presented data are easier to read and understand if the text is aligned to left

Table 1: the presented data are easier to read and understand if the text is aligned to left in each column.

I read carefully the paper and the supplementary materials; I do not have any other comments. I asked for minor revision, mainly for the formatting suggestions for Table 1 and Table 3.

Author Response

Below we provide a point-by-point response to reviewer 2. In addition, attached you will find a document containing the responses to all reviewers:

Reviewer 2

The paper meets all of the requirements of an “evidence map” as the specified protocol, a systematic search strategy, a systematically applied, clear inclusion and exclusion criteria, and reports all eligible studies (in the supplementary file S4).

Being in the third year of the COVID-19 pandemic (when vaccines are available, changing almost fundamentally the situation along with the healthcare needs, as it is pointed out among the limitations of the study) the importance of the presented data regarding the first year of the pandemic is decreasing, but the work still has a significant contribution to the field.

Some of the identified needs and gaps are no longer valid; there are needs, which are not relevant anymore (personal protective equipment are available now, a wider range of knowledge about the virus is available) and other still stands, but in the chapter 4.3. (Implications for practice) the valid interventional proposals are formulated.

It is a thoroughly documented, very well-written paper.

We understand the reviewer’s concern and have revised the discussion accordingly. We explicitly highlight that whereas some of the identified needs and gaps have largely been addressed (e.g., PPE), others remain in force two years after the onset of the pandemic. More specifically, we added the following to section 4.4:

“some of the needs and gaps identified during the first year of the pandemic have already largely been addressed, such as the need for adequate supply of appropriate PPE for professionals (Haegdorens et al., 2022). Nevertheless, this information will be useful in facilitating the preparation of health systems for future pandemics. Other identified needs and gaps, however, are still current and pressing, such as the need to improve access to mental health professionals and efforts should be undertaken to prioritize and address them.”

Row 161: “see also 3” – is not obvious what it refers to

We have specified what we refer to. The sentence now reads,

“Family” is defined as any group of persons who are related biologically, emotionally, or legally such as siblings, parents, spouses, hired caregivers, significant others, and friends (see Omole et al., 2011 for a similar definition of “family” [41]).

Table 1: the presented data are easier to read and understand if the text is aligned to left in each column.

Done.

Table 3 – bullets are not necessary, and the presented data are easier to read and understand if the text is aligned to left

Done. The bullet points have been removed and the text has been aligned to the left.

I read carefully the paper and the supplementary materials; I do not have any other comments. I asked for minor revision, mainly for the formatting suggestions for Table 1 and Table 3.

Reviewer 3 Report

Comments on “Healthcare Needs Related to COVID-19: An Evidence  Map of the First Year of the Pandemic”

Comments:

1.This is a well written examination of global health care needs reviewing the first year of the COVID-19 pandemic.  It presents a well-researched paper highlighting key concerns of both health providers and patients from a variety of countries.  It gives evidence through in-depth studies of the challenges for both groups from a number of countries. It provides extensive documentation to support the conclusions. The visuals are very useful to navigate the relationship of the evidence to the conclusions. The evidence provided is very well documented. The strengths and limitations of the study are clearly defined.  I do have the comments below which I believe would strengthen the paper.

2. It would highlight the are of research to include in the title the  breath of the investigation.  The title might read “Healthcare Needs Related to COVID-19: An Evidence Map of the First Year of the Pandemic:  A Global perspective.

3. For readers unfamiliar with Evidence Map, a more thorough presentation of this approach would be most helpful.  What is its history? Where has it been used before? In addition, presenting the framework of this approach would help readers follow the arguments.

4. In the section entitled Results, it would be more appropriate to place points 3.1-3.3 in the section entitled  Materials and Methods. 3.3.1and all information that follows in this section should stay in Results.

5.  The supplemental material are key to supporting the arguments. It would be most helpful to put these in the Appendix rather than supplemental files.  Readers could more easily access the evidence for the arguments.

6. The key words are weak.  They should give the reader more information about what to expect in this paper.

Author Response

Below we provide a point-by-point response to reviewer 3. In addition, attached you will find a document containing the responses to all reviewers:

Reviewer 3 

This is a well written examination of global health care needs reviewing the first year of the COVID-19 pandemic.  It presents a well-researched paper highlighting key concerns of both health providers and patients from a variety of countries.  It gives evidence through in-depth studies of the challenges for both groups from a number of countries. It provides extensive documentation to support the conclusions. The visuals are very useful to navigate the relationship of the evidence to the conclusions. The evidence provided is very well documented. The strengths and limitations of the study are clearly defined.  I do have the comments below which I believe would strengthen the paper.

I would highlight the area of research to include in the title the  breath of the investigation.  The title might read “Healthcare Needs Related to COVID-19: An Evidence Map of the First Year of the Pandemic:  A Global perspective.

We agree this could be a nice change.  The proposed new title reads, “Global Healthcare Needs Related to COVID-19: An Evidence Map of the First Year of the Pandemic”.

For readers unfamiliar with Evidence Map, a more thorough presentation of this approach would be most helpful.  What is its history? Where has it been used before? In addition, presenting the framework of this approach would help readers follow the arguments.

Done. We have provided a more thorough description of Evidence Mapping in section 1.1. This section now reads,

“Evidence mapping is an emerging method for reviewing and synthetizing evidence in a reproducible and transparent manner (CEE, 2013). It was first developed and used in the social sciences (Bates et al., 2007; Clapton et al., 2009), and it has since been widely used in other fields including health sciences (e.g., patient preferences, Gonzalez et al., 2019; central nervous system injury, Bragge et al. 2011; environmental management, Haddaway et al., 2016). More specifically, evidence mapping is a systematic search of a broad subject area used to identify clusters and gaps in knowledge and/or future research needs that presents results in a user-friendly product, such as a visual or a searchable database (Miake-Lye et al., 2016). This framework does not require critical appraisal of included studies, but instead aims to assess what type of research has been conducted, what settings have been evaluated, and what methods have been used. Qualitative or quantitative analyses are not always performed; alternatively, the synthesis is limited to a narrative description of the state of knowledge (Haddaway et al., 2016). In sum, evidence mapping is particularly useful as an exploratory, comprehensive evidence synthesis method of a broad topic to provide an overview of the evidence base in an easily digestible format.”

In the section entitled Results, it would be more appropriate to place points 3.1-3.3 in the section entitled  Materials and Methods. 3.3.1and all information that follows in this section should stay in Results.

Based on the PRISMA extension for scoping reviews (PRISMA-ScR), the selection of sources of evidence and the main characteristics of the included studies must be placed in the Results section. Therefore, in compliance with the PRISMA statement, we have decided to keep points 3.1-3.3 in the section entitled “Results”.

The supplemental material are key to supporting the arguments. It would be most helpful to put these in the Appendix rather than supplemental files.  Readers could more easily access the evidence for the arguments.

Thank you for your comment. The importance of the supplemental materials in supporting the arguments is well understood. However, since the supplemental files would add considerable length to an already lengthy manuscript, and given the concern expressed by another reviewer regarding the length of the manuscript, we have kept these files as supplemental material. 

The key words are weak.  They should give the reader more information about what to expect in this paper.

We have added the key words in bold accordingly, as follows:

COVID-19; pandemic; needs assessment; healthcare needs; healthcare professionals' needs; patients' needs; family members' needs; evidence map; systematic review.

We welcome specific suggestions to further improve the key words if needed.

Round 2

Reviewer 1 Report

Very nice piece of work, still very lengthy. 

No other comments.

Reviewer 3 Report

The authors have addressed my concerns.  The paper is now ready for publication.